# Co-Tuning for Transfer Learning

**Kaichao You, Zhi Kou, Mingsheng Long** (✉)**, Jianmin Wang**
School of Software, BNRist, Research Center for Big Data, Tsinghua University, China
{ykc20,kz19}@mails.tsinghua.edu.cn, {mingsheng,jimwang}@tsinghua.edu.cn

## Abstract

Fine-tuning pre-trained deep neural networks (DNNs) to a target dataset, also known as transfer learning, is widely used in computer vision and NLP. Because task-specific layers mainly contain categorical information and categories vary with datasets, practitioners only *partially* transfer pre-trained models by discarding task-specific layers and fine-tuning bottom layers. However, it is a reckless loss to simply discard task-specific parameters which take up as many as 20% of the total parameters in pre-trained models. To *fully* transfer pre-trained models, we propose a two-step framework named **Co-Tuning**: (i) learn the relationship between source categories and target categories from the pre-trained model with calibrated predictions; (ii) target labels (one-hot labels), as well as source labels (probabilistic labels) translated by the category relationship, collaboratively supervise the fine-tuning process. A simple instantiation of the framework shows strong empirical results in four visual classification tasks and one NLP classification task, bringing up to 20% relative improvement. While state-of-the-art fine-tuning techniques mainly focus on how to impose regularization when data are not abundant, Co-Tuning works not only in medium-scale datasets (100 samples per class) but also in large-scale datasets (1000 samples per class) where regularization-based methods bring no gains over the vanilla fine-tuning. Co-Tuning relies on a typically valid assumption that the pre-trained dataset is diverse enough, implying its broad application areas.

## 1 Introduction

The practice of fine-tuning pre-trained models, also known as transfer learning, is prevalent in deep learning. In computer vision, we have models pre-trained on the ImageNet (Deng et al., 2009) classification task, such as ResNet (He et al., 2016) and DenseNet (Huang et al., 2017). In NLP, we witness the recent emergence of self-attention (Vaswani et al., 2017) based models pre-trained on a large-scale corpus for masked language modeling, including BERT (Devlin et al., 2019), XLNet (Yang et al., 2019), etc. Compared with training from scratch, fine-tuning requires less labeled data, enables faster training, and usually achieves better performance (He et al., 2019).

The rule of thumb for many deep learning tasks is to (i) start from a pre-trained model, (ii) remove task-specific top layers, and (iii) fine-tune bottom layers on the target task as a feature extractor. This way, pre-trained models are *partially transferred* because only parameters in bottom layers are transferred. As shown in Table 1, however, parameters in task-specific layers can take up over 20% of the total parameters. Simply discarding these layers is a reckless waste. To make the best of pre-trained models, task-specific layers should also be transferred to achieve *full transfer* of DNNs.

Although full transfer of DNNs is appealing, an obvious obstacle of transferring task-specific layers is that categories vary with datasets and it is difficult to reuse these layers. For example, when transferring ImageNet pre-trained models to COCO-70 [1], the relationship between 1000 categories

Table 1: Parameter count in popular pre-trained models from `torchvision` and `transformers`.

| Pre-trained model | ResNet-50 | DenseNet-121 | Inception-V3 | BERT-base |
|---|---|---|---|---|
| Task-specific parameters / Million | 2.0 | 1.0 | 2.0 | 22.9 |
| Total parameters / Million | 25.6 | 8.0 | 27.2 | 108.9 |
| Percentage / % | 7.8 | 12.5 | 7.4 | **21.0** |

in ImageNet and 70 categories in COCO-70 is required to transfer task-specific layers. However, manually figuring out the relationship has many problems, including:

- *Same concept, different names*. COCO concept "elephant" almost means the same as ImageNet concepts "indian elephant" and "african elephant", but with different names.
- *Same name, different concepts*. The concept "horse" in COCO often co-occurs with human, while the concept "horse" in ImageNet often co-occurs with cart. Although they share the same name "horse", they have different contexts.
- *Probabilistic relationship*. Concept relationship is probabilistic rather than binary, which is hard for human to reason about (see Fig. 1 for example).

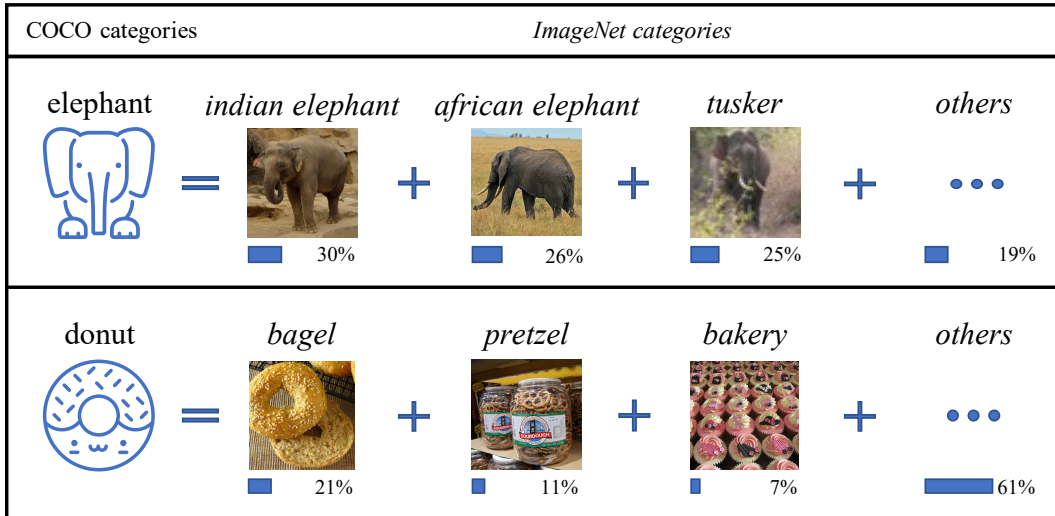

Figure 1: The relationship between COCO categories and ImageNet categories learned by Alg. 1. The algorithm successfully figures out the relationship among both concrete categories ("elephant", "tusker", "donut", "bagel") and abstract categories ("bakery").

To address the problem, we propose a **Co-Tuning** framework for transfer learning to achieve full transfer of pre-trained DNNs. It first learns the relationship between source categories and target categories (see Fig. 1 as an example), then translates one-hot target labels into probabilistic source labels which collaboratively supervise the fine-tuning process. We also propose two possible approaches to learn the category relationship. It is also highly pluggable and easy to use.

The Co-Tuning framework is empirically evaluated in four visual classification tasks and one named entity recognition task. It brings unanimous improvement compared to standard fine-tuning, with **up to** 20% **improvement**. When abundant data are available and regularization-based methods fail to improve over the vanilla fine-tuning, Co-Tuning still brings gains by a large margin.

## 2 Related Work

Below we review pre-training and transfer learning of deep neural networks, respectively. Some other related topics are also included for a complete discussion.

## 2.1 Pre-training of Deep Neural Networks

In computer vision, AlexNet (Krizhevsky et al., 2012) is a seminal deep learning model to surpass feature-engineering methods in the ImageNet classification task. To further improve the performance, researchers have explored different network designs. VGG (Simonyan & Zisserman, 2015) increases the depth of neural networks by carefully designing the architecture and the training scheme. Highway Networks (Srivastava et al., 2015) create gradient highway to ease training. ResNet (He et al., 2016) introduces skip connections and DenseNet (Huang et al., 2017) exploits dense connections.

Pre-trained DNNs have also achieved remarkable advances in NLP. Word2Vec (Mikolov et al., 2013) and Glove (Pennington et al., 2014) provide pre-trained word representations, while downstream tasks have to learn high-level semantic representations from scratch. BERT (Devlin et al., 2019), which is trained by masked language modeling, gives powerful pre-trained representation that benefits a variety of downstream tasks by simple fine-tuning. After the emergence of BERT, fine-tuning pre-trained DNNs becomes popular in NLP and is the state-of-the-art for many tasks.

The advances of pre-trained models greatly benefit the research community. Kornblith et al. (2019) empirically show that better pre-trained models tend to have better transfer learning performance.

## 2.2 Transfer Learning of Deep Neural Networks

Donahue et al. (2014); Oquab et al. (2014) confirm that features extracted by pre-trained AlexNet can be transferred to various tasks and work better than hand-crafted features. Yosinski et al. (2014); Agrawal et al. (2014); Girshick et al. (2014) later show that fine-tuning pre-trained networks provides better performance than fixed pre-trained representations. Even when the target dataset is very dissimilar to the pre-trained dataset and fine-tuning brings no performance gain (Raghu et al., 2019), it can accelerate the convergence speed (He et al., 2019).

Recent works on fine-tuning mainly focus on how to better exploit the inductive bias of pre-trained models, i.e., how to regularize fine-tuning with pre-trained models. L2-SP (Li et al., 2018) regularizes weights to be close to the pre-trained weights by imposing L2 constraint, exploiting the inductive bias of the "mechanism" (weight) of neural networks. DELTA (Li et al., 2019) computes channel-wise importance of pre-trained networks and penalizes network activations that deviate from pre-trained activations, exploiting the inductive bias of the "behavior" (activation) of neural networks. BSS (Chen et al., 2019) penalizes small eigenvalues of learned representation because they find small eigenvalues cause negative transfer after empirically analyzing the eigenvalue spectrum of learned representations. NLP research on fine-tuning has an alternative focus on resource consumption (Houlsby et al., 2019; Garg et al., 2020) and selective layer freezing (Sun et al., 2019; Wang et al., 2019).

Although there exist some techniques to improve the vanilla fine-tuning, they are not widely adopted because of unsatisfactory performance or complex design. Kornblith et al. (2019) launch a systematic investigation of fine-tuning with grid search of the hyper-parameters. Li et al. (2020) spend about 10K GPU hours on tuning hyper-parameters of fine-tuning and provide some practical guidelines. These recent large-scale empirical analyses of fine-tuning focus on vanilla fine-tuning: replace the top layer(s) with a randomly initialized fully-connected layer and train the network in a supervised way. In addition, Li et al. (2020) systematically study the effect of L2-SP and find that the regularization-based method does not work well if the target dataset is dissimilar to the pre-trained dataset. Practitioners still stick to vanilla fine-tuning because yet there are no *practical* transfer learning algorithms that consistently outperform the vanilla fine-tuning.

This paper presents Co-Tuning, an algorithm framework to fully transfer pre-trained models. It outperforms vanilla fine-tuning by a large margin, is easy to use and can be of interest to practitioners.

## 2.3 Other Related Topics

**Domain adaptation** (Long et al., 2015; Ganin & Lempitsky, 2015; Saito et al., 2018) aims to improve the performance in the target domain by leveraging source data. In domain adaptation, category spaces of both domains are the same, and both target data (weakly labeled or unlabeled) and source data (fully labeled) are required during training. In transfer learning (fine-tuning), the category spaces of both datasets can differ; and only target data and pre-trained models are available for training.

**Continual Learning** aims at learning new knowledge without forgetting old knowledge, i.e. avoiding catastrophic forgetting. Kirkpatrick et al. (2017) propose elastic weight consolidation to elastically restrict the weight change. Li & Hoiem (2018) propose Learning without Forgetting to force the prediction on old tasks to be similar to the output of models trained in old tasks. Although they work well in continual learning, their results are not advantageous in transfer learning. Experiments show that Co-Tuning fits much better for transfer learning because it explicitly models the relationship between categories and exploits the beneficial relationship to co-tune the target network.

## 3 Transfer Learning Setup and Existing Solutions

Given a DNN $f_0$ [2] pre-trained on a source dataset $\mathcal{D}_s = \{(x_s^i, y_s^i)\}_{i=1}^{m_s}$, transfer learning aims to fine-tune $f_0$ to a target dataset $\mathcal{D}_t = \{(x_t^i, y_t^i)\}_{i=1}^{m_t}$. $\mathcal{D}_s$ and $\mathcal{D}_t$ share the same input space $\mathcal{X}$ but have respective category spaces $\mathcal{Y}_s$ and $\mathcal{Y}_t$. Because $\mathcal{D}_s$ is often large-scale and lacks portability, only $\mathcal{D}_t$ and $f_0$ pre-trained on $\mathcal{D}_s$ are available during fine-tuning. In computer vision, typically $\mathcal{D}_s$ denotes ImageNet and $\mathcal{D}_t$ is the visual classification dataset we concern. Two datasets share the image space as input but have different categories as output spaces.

Because $\mathcal{Y}_s$ and $\mathcal{Y}_t$ are heterogeneous, $f_0$ pre-trained on $\mathcal{D}_s$ cannot be directly applied to target data. A common practice is to split $f_0$ into two parts: a general representation function $F_{\bar{\theta}^0}$ (parametrized by $\bar{\theta}^0$) and a task-specific function $G_{\theta_s^0}$ (parametrized by $\theta_s^0$) which denotes the top layers of pre-trained model. Then the representation function is retained and the task-specific function is replaced by a randomly initialized function $H_{\theta_t}$ whose output space matches $\mathcal{Y}_t$. This gives rise to the vanilla fine-tuning method by solving Eq. 1, where $\ell(\cdot, \cdot)$ is a loss function and cross-entropy is a typical choice for classification. Initialization matters in optimizing DNNs because it is a non-convex optimization problem. Pre-trained parameters $\bar{\theta}^0$ provide a good starting point for the optimization:

$$(\bar{\theta}^*, \theta_t^*) = \arg\min_{\bar{\theta}, \theta_t} \frac{1}{|\mathcal{D}_t|} \sum_{i=1}^{m_t} \ell(H_{\theta_t}(F_{\bar{\theta}}(x_t^i)), y_t^i). \tag{1}$$

Besides vanilla fine-tuning, there are also techniques focusing on preventing over-fitting. They follow a structural risk minimization framework with various regularizers. Li et al. (2018) adopt $||\bar{\theta} - \bar{\theta}^0||_2^2$ to constrain the weight $\bar{\theta}$ to be near its starting point $\bar{\theta}^0$. Li et al. (2019) adopt $= ||F_{\bar{\theta}}(x_t^i) - F_{\bar{\theta}^0}(x_t^i)||_2^2$ with some pre-computed channel-wise attention weights to regularize the activation of networks during fine-tuning. Chen et al. (2019) penalize small eigenvalues of activations in each mini-batch.

Existing solutions to transfer learning all discard task-specific function $G_{\theta_s^0}$, recklessly wasting the parameters and the categorical knowledge therein, even though $G_{\theta_s^0}$ contains a decent proportion of parameters. To fully transfer pre-trained DNNs, next we propose Co-Tuning for transfer learning.

## 4 Co-Tuning for Transfer Learning

### 4.1 Co-Tuning Framework

Since the output of task-specific function $G_{\theta_s^0}$ is a probability distribution over source categories $\mathcal{Y}_s$ and only target labels are available during fine-tuning, we have to figure out firstly the relationship between category spaces, i.e., the conditional distribution $p(y_s|y_t)$.

If the category relationship $p(y_s|y_t)$ is known, we can translate target label $y_t$ into probabilistic source categories $y_s$, which can be used to fine-tune the task-specific function $G_{\theta_s}$. The gradient of $G_{\theta_s}$ can be back-propagated into $F_{\bar{\theta}}$, bringing in additional supervision. As described in Eq. 2 and Fig. 2, $y_t$ and $y_s$ collaboratively supervise the transfer learning process, where $\lambda$ trades off the target supervision and source supervision, $\bar{\theta}, \theta_s$ are initialized from pre-trained weight $\bar{\theta}^0, \theta_s^0$. In this way, we can fully exploit pre-trained parameters $\bar{\theta}^0, \theta_s^0$ in a collaborative training style. When the training finishes, $G_{\theta_s}$ can be removed so that fine-tuning is improved *without* additional inference cost.

$$(\bar{\theta}^*, \theta_t^*, \theta_s^*) = \arg\min_{\bar{\theta}, \theta_t, \theta_s} \frac{1}{|\mathcal{D}_t|} \sum_{i=1}^{m_t} \left[ \ell(H_{\theta_t}(F_{\bar{\theta}}(x_t^i)), y_t^i) + \lambda\, \ell(G_{\theta_s}(F_{\bar{\theta}}(x_t^i)), p(y_s|y_t = y_t^i)) \right]. \tag{2}$$

We dub this framework "Co-Tuning" because it uses both the ground-truth $y_t$ and probabilistic $y_s$ estimated from the category relationship to fine-tune the full pre-trained network. It works by bringing in additional supervision to fine-tuning, and is easy to implement based on Eq. 2.

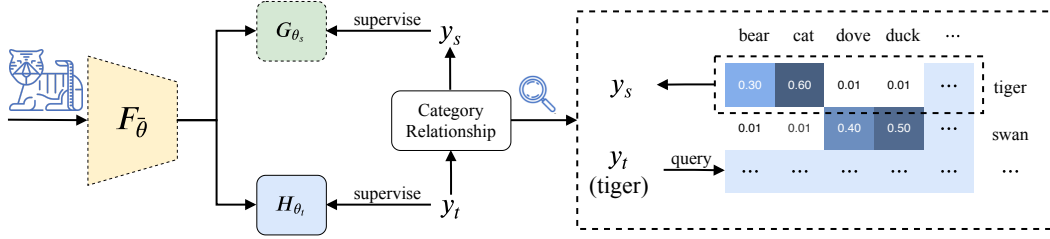

Figure 2: Training pipeline of Co-Tuning.

## 4.2 Learning Category Relationship

The only unresolved problem in the Co-Tuning framework is how to learn the category relationship $p(y_s|y_t)$. Below we propose two ways to learn $p(y_s|y_t)$.

A direct approach (Eq. 3) is to average the predictions of the pre-trained model over all samples of each target category, where the pre-trained model $f_0$ is regarded as a probabilistic model approximating the conditional distribution $f_0(x) \approx p(y_s|x)$.

$$p(y_s|y_t = y) \approx |\mathcal{D}_t^y|^{-1} \Sigma_{(x,y_t) \in \mathcal{D}_t^y} f_0(x), \quad \mathcal{D}_t^y = \{(x, y_t) \in \mathcal{D}_t | y_t = y\}. \quad (3)$$

Because categories in the pre-trained dataset are diverse enough to serve as basic categories to compose target category (as shown in Fig. 1), a reverse approach is to first learn the mapping $y_s \rightarrow y_t$ from $(f_0(x_t), y_t)$ pairs, where $y_t$ is target label and $f_0(x) \approx p(y_s|x)$ is a probability distribution over source categories $\mathcal{Y}_s$. Then $p(y_s|y_t)$ can be computed from $p(y_t|y_s)$ by Bayes's rule.

In practice, the direct approach is simple and straightforward, while the reverse one is more effective. Therefore, we mainly study the reverse approach in our experiments, but also report the direct approach in ablation study. Alg. 1 describes the full procedure of the reverse approach we propose.

---

**Algorithm 1** Category relationship learning (the reverse approach)

---

**Input:** $f_0$, source validation data $\mathcal{D}_s^v = \{(x_s^i, y_s^i)\}_{i=1}^{m_v}$, target training data $\mathcal{D}_t = \{(x_t^i, y_t^i)\}_{i=1}^{m_t}$
**Output:** Category relationship $p(y_s|y_t)$
Call Alg. 2 to calibrate $f_0$ with $\mathcal{D}_s^v$, which returns the calibrated deep model $\tilde{f}_0$
Construct $\tilde{\mathcal{D}}_t = \{(\tilde{f}_0(x_t^i), y_t^i)\}_{i=1}^{m_t}$, further split it into training set $\tilde{\mathcal{D}}_t^{train}$ and validation set $\tilde{\mathcal{D}}_t^v$
Learn a neural network $g$ from $\tilde{\mathcal{D}}_t^{train}$ to map calibrated source predictions to target labels
Call Alg. 2 to calibrate $g$ with $\tilde{\mathcal{D}}_t^v$, which returns $p(y_t|y_s) \approx \tilde{g}(y_s)$
Compute marginal probability $p(y_s)$ and $p(y_t)$ from $\tilde{\mathcal{D}}_t$
Compute $p(y_s|y_t)$ by Bayes's rule: $p(y_s = i|y_t = j) = \frac{p(y_s=i)}{p(y_t=j)}p(y_t = j|y_s = i)$
Return $p(y_s|y_t)$

---

## 4.3 Neural Network Calibration

Alg. 1 mentions a calibration procedure because we want $f_0(x)$ to reflect the probability of source categories with high fidelity. Without calibration, DNNs can be over-confident (Guo et al., 2017). Therefore, we have to calibrate the probability output of $f_0(x)$. Guo et al. (2017) show that neural networks calibration can be done by minimizing negative log-likelihood (NLL) on validation data through adjusting a single temperature (Hinton et al., 2015) parameter. The formulation leads to a convex optimization problem which is ready to solve. Alg. 2 describes the calibration procedure.

Note that the calibration of $f_0$ only depends on the pre-trained dataset and the pre-trained model. *We advocate that pre-trained model providers release pre-trained models and their calibrated version,*

---
**Algorithm 2** Neural network calibration
---
**Input:** DNN $f$ that outputs uncalibrated logits, validation data $\mathcal{D} = \left\{ x^i, y^i \right\}_{i=1}^m$

**Output:** A neural network $\tilde{f}$ that outputs **calibrated** logits

Compute the scaling parameter $t^* = \arg\min_{t>0} \sum_{i=1}^m \texttt{cross\_entropy}(\texttt{softmax}(f(x^i)/t), y^i)$

Return $\tilde{f}$, where $\tilde{f}(x) = f(x)/t^*$
---

which only requires an additional scalar parameter. Given the calibrated version, pre-trained source data is not necessary in this regard.

To summarize, we propose the Co-Tuning framework to fully transfer pre-trained models, as well as two approaches to learn the category relationship $p(y_s|y_t)$. We also introduce the calibration option if source validation data or the confidence-calibrated version of pretrained model is readily available.

## 5   Experiments

In this section, we empirically evaluate Co-Tuning in several dimensions:

- Task: 4 visual classification tasks and one NLP task (named entity recognition).
- Dataset scale: medium-scale dataset ($\approx$ 100 samples per class) and large-scale dataset ($\approx$ 1000 samples per class). We also explore different sampling rates (the proportion of images used for training) to compare the performance among a wide spectrum of dataset scales.
- Pre-trained model: ResNet-50, DenseNet-121 and BERT-base.

Co-Tuning is compared against three state-of-the-art fine-tuning methods: $L^2$-SP (Li et al., 2018), DELTA (Li et al., 2019) and BSS (Chen et al., 2019). Vanilla fine-tuning serves as a strong baseline. Experiments confirm that Co-Tuning brings unanimous improvement across all tasks, dataset scales and pre-trained models. By "Co-Tuning", we mean the instantiation of the Co-Tuning framework with calibration option and $p(y_s|y_t)$ learned by the reverse approach. At last, we present analyses and ablation studies of each component in Co-Tuning.

### 5.1   Implementation Details

Basically, we follow the common practice in the community as described in Chen et al. (2019). The partition of task-specific layers $G_{\theta_s}$ and bottom layers $F_{\bar{\theta}}$ follows official transfer learning guidelines of `torchvision` and `transformers`: for ResNet-50 and DenseNet-121, $G_{\theta_s}$ means the last fully-connected layer; for BERT-base, $G_{\theta_s}$ means the last classification block; $F_{\bar{\theta}}$ are the rest layers. Target layer $H_{\theta_t}$ is a fully-connected layer with randomly initialized parameters. The learning rate for randomly initialized parameters is ten times of the learning rate for pre-trained parameters, following the *de facto* practice of Yosinski et al. (2014). Hyper-parameters of Co-Tuning and compared methods are selected by the performance on target validation data. Validation data of pre-trained datasets are used to calibrate pre-trained models. We use softmax regression provided by `scikit-learn` (Pedregosa et al., 2011) for the neural network $g$ in Alg. 1, and try other choices in ablation study. We choose PyTorch (Benoit et al., 2019) deep learning framework for its flexibility. Unless stated, all models are optimized by SGD with 0.9 momentum. Each experiment is repeated three times with different random seeds to collect mean and standard deviation of the performance.

The details of evaluation and train/val/test split are as follows. Each dataset has a train/test split. For datasets without validation splits, 20% training data are used for validation (split once and then the validation set is fixed) and the rest 80% training data are used for training. This way, each dataset has a train/val/test split. Sampling rate further determines the actually used training data. For example, 30% sampling rate means to use 30% training data for training, leaving the rest 70% training data untouched (both labels and image/text). Each method has access to the same set of training data. In Alg. 1, Co-Tuning further splits the sampled training data into secondary train/val (Alg. 1, Line 4) to learn the category relationship. This way, the empirical evaluation is fair because not any method uses extra data/labels compared to others.

Code is available at `https://github.com/thuml/CoTuning`.

## 5.2 Medium-scale Classification

For medium-scale classification tasks, we use CUB-200-2011 (Welinder et al., 2010), Stanford Cars (Krause et al., 2013), and FGVC Aircraft (Maji et al., 2013) datasets. The numbers of images and classes are: CUB-200-2011 (11,788 images for 200 bird species), Stanford Cars (16,185 images for 196 car categories), and FGVC Aircraft (10,000 images for 100 aircraft variants). They have about 100 images per class. These medium-scale fine-grained classification datasets are extensively investigated in transfer learning (Chen et al., 2019; Li et al., 2018, 2019, 2020).

Results are reported in Table 2, with pre-trained ResNet-50. Across all of the sampling rates and all of the datasets, **Co-Tuning performs significantly better than vanilla fine-tuning, L2-SP, DELTA, and BSS**. All of the best results (in bold) are achieved by Co-Tuning. In Stanford Cars dataset with sampling rate of 15%, baseline fine-tuning accuracy is 36.77% and Co-Tuning improves the accuracy to 46.02%, bringing in up to **20**% relative improvement.

It is worth to note that Co-Tuning is pretty robust to hyper-parameters: $\lambda = 2.3$ cross-validated on one task works well for these three datasets and four sampling rates.

Table 2: Classification accuracy in medium-scale classification datasets (Pre-trained ResNet-50).

| Dataset | Method | Sampling Rates | | | |
|---|---|---|---|---|---|
| | | 15% | 30% | 50% | 100% |
| CUB-200-2011 | Fine-tune (baseline) | $45.25 \pm 0.12$ | $59.68 \pm 0.21$ | $70.12 \pm 0.29$ | $78.01 \pm 0.16$ |
| | L$^2$-SP (Li et al., 2018) | $45.08 \pm 0.19$ | $57.78 \pm 0.24$ | $69.47 \pm 0.29$ | $78.44 \pm 0.17$ |
| | DELTA (Li et al., 2019) | $46.83 \pm 0.21$ | $60.37 \pm 0.25$ | $71.38 \pm 0.20$ | $78.63 \pm 0.18$ |
| | BSS (Chen et al., 2019) | $47.74 \pm 0.23$ | $63.38 \pm 0.29$ | $72.56 \pm 0.17$ | $78.85 \pm 0.31$ |
| | Co-Tuning | $\mathbf{52.58} \pm 0.53$ | $\mathbf{66.47} \pm 0.17$ | $\mathbf{74.64} \pm 0.36$ | $\mathbf{81.24} \pm 0.14$ |
| Stanford Cars | Fine-tune (baseline) | $36.77 \pm 0.12$ | $60.63 \pm 0.18$ | $75.10 \pm 0.21$ | $87.20 \pm 0.19$ |
| | L$^2$-SP (Li et al., 2018) | $36.10 \pm 0.30$ | $60.30 \pm 0.28$ | $75.48 \pm 0.22$ | $86.58 \pm 0.26$ |
| | DELTA (Li et al., 2019) | $39.37 \pm 0.34$ | $63.28 \pm 0.27$ | $76.53 \pm 0.24$ | $86.32 \pm 0.20$ |
| | BSS (Chen et al., 2019) | $40.57 \pm 0.12$ | $64.13 \pm 0.18$ | $76.78 \pm 0.21$ | $87.63 \pm 0.27$ |
| | Co-Tuning | $\mathbf{46.02} \pm 0.18$ | $\mathbf{69.09} \pm 0.10$ | $\mathbf{80.66} \pm 0.25$ | $\mathbf{89.53} \pm 0.09$ |
| FGVC Aircraft | Fine-tune (baseline) | $39.57 \pm 0.20$ | $57.46 \pm 0.12$ | $67.93 \pm 0.28$ | $81.13 \pm 0.21$ |
| | L$^2$-SP (Li et al., 2018) | $39.27 \pm 0.24$ | $57.12 \pm 0.27$ | $67.46 \pm 0.26$ | $80.98 \pm 0.29$ |
| | DELTA (Li et al., 2019) | $42.16 \pm 0.21$ | $58.60 \pm 0.29$ | $68.51 \pm 0.25$ | $80.44 \pm 0.20$ |
| | BSS (Chen et al., 2019) | $40.41 \pm 0.12$ | $59.23 \pm 0.31$ | $69.19 \pm 0.13$ | $81.48 \pm 0.18$ |
| | Co-Tuning | $\mathbf{44.09} \pm 0.67$ | $\mathbf{61.65} \pm 0.32$ | $\mathbf{72.73} \pm 0.08$ | $\mathbf{83.87} \pm 0.09$ |

## 5.3 Large-scale Classification

Previous works focus on regularization and only experiment with medium-scale datasets. We further explore transfer learning with a constructed large-scale dataset in this section.

The large-scale dataset is constructed from COCO object detection task in 2017. For each object annotation, we crop the bounding box with paddings. For example, if the bounding box for an object is $50 \times 50$ pixels, we crop $70 \times 70$ region, with the bounding box in the center. The padding is necessary because we often crop a smaller region of images for training computer vision models. Small objects (both width and height are smaller than 50 pixels) are removed. Originally there are 80 categories (not including the background category) in COCO object detection task. To explore fine-tuning in a large-scale dataset, we only keep categories with more than 1000 images, resulting in 70 categories. The constructed dataset is named COCO-70.

In COCO-70, we compare Co-Tuning with other fine-tuning methods across different sampling rates. To avoid over-depending on a specific pre-trained model, we switch to pre-trained DenseNet-121. Results are presented in Table 3. It is clear that, as there are more and more data (sampling rate goes from 15% to 100%), the efficacy of regularization-based methods (L2-SP, DELTA and BSS) fades away quickly. With 100% data, their improvement is within standard deviation. On the contrary, Co-Tuning shows consistent gains by large margins even when there are abundant labeled data.

The results are not surprising, though. Regularization reflects our prior knowledge on what is a good model (Bishop, 2006) and mainly works when data are limited. As more and more data are available, regularization typically does not help. On the contrary, Co-Tuning brings in additional supervision by

exploiting the category relationship and task-specific layers of pre-trained networks. The effect of additional supervision will not fade away in general even when there are abundant data.

Table 3: Classification accuracy in large-scale COCO-70 dataset (Pre-trained DenseNet-121).

| Method | Sampling Rates | | | |
|---|---|---|---|---|
| | 15% | 30% | 50% | 100% |
| Fine-tune (baseline) | $76.60 \pm 0.04$ | $80.15 \pm 0.25$ | $82.50 \pm 0.43$ | $84.41 \pm 0.22$ |
| $L^2$-SP (Li et al., 2018) | $77.53 \pm 0.47$ | $80.67 \pm 0.29$ | $83.07 \pm 0.39$ | $84.78 \pm 0.16$ |
| DELTA (Li et al., 2019) | $76.94 \pm 0.37$ | $79.72 \pm 0.24$ | $82.00 \pm 0.52$ | $84.66 \pm 0.08$ |
| BSS (Chen et al., 2019) | $77.39 \pm 0.15$ | $80.74 \pm 0.22$ | $82.75 \pm 0.59$ | $84.71 \pm 0.13$ |
| Co-Tuning | $\mathbf{77.64} \pm 0.23$ | $\mathbf{81.19} \pm 0.18$ | $83.43 \pm 0.22$ | $\mathbf{85.65} \pm 0.11$ |

## 5.4 Named Entity Recognition

Co-tuning can also be extended to NLP tasks. We experiment with English named entity recognition (NER) task in CoNLL 2003 (Sang & De Meulder, 2003). NER is a token-level classification problem which requires classification of each token (word). The performance of NER is measured by the F1-score of named entities. Since fine-tuning BERT (Devlin et al., 2019) is overwhelmingly used in NLP, we modify the NER transfer learning script in `transformers`, fine-tuning pre-trained BERT-base model with Co-Tuning by Adam (Kingma & Ba, 2015). Note that BERT has 2 pre-trained tasks: masked language modeling (where the target label space is the vocabulary) and next sentence prediction (where the target label space is binary), we only use the masked language modeling part. Because the pre-trained dataset for BERT is too large, we do not include calibration option in this task. The vanilla fine-tuning baseline achieves F1-score of 90.81, BSS and L2-SP achieve 90.85 and 91.02 respectively. Co-Tuning improves vanilla fine-tuning to 91.27.

## 5.5 Ablation Study

We conduct ablation study in Table 4 to explore the effect of each component in Co-Tuning. Results are calculated on CUB dataset with 15% data. Co-Tuning and vanilla fine-tuning are listed for comparison. Co-Tuning (T) is the variant that uses a two-layer neural network as the function $g$ in Alg. 1 to replace the softmax regression. Co-Tuning (D) is the variant that uses the direct approach for learning $p(y_s|y_t)$. Co-Tuning w/o calibration is the variant that removes the calibration steps in Alg. 1. In addition, we compare Co-Tuning with several intuitive ways of re-using task-specific parameters:

Table 4: Ablation study in CUB with 15% data.

| Method | Accuracy (%) |
|---|---|
| Co-Tuning | $52.58 \pm 0.53$ |
| Co-Tuning (T) | $52.03 \pm 0.21$ |
| Co-Tuning (N) | $51.20 \pm 0.28$ |
| Co-Tuning w/o calibration | $50.67 \pm 0.41$ |
| LwF (Li & Hoiem, 2018) | $47.67 \pm 0.37$ |
| Fine-tune (baseline) | $45.25 \pm 0.12$ |
| AddLayer | $42.68 \pm 0.37$ |

(i) adding a new lay on top of the whole pre-trained model (AddLayer); (ii) Learning without Forgetting (LwF) (Li & Hoiem, 2018) designed for continual learning. From the ablation study, we conclude that: one-layer softmax regression is enough for $g$; the direct approach of learning $p(y_s|y_t)$ also works well, but is inferior to the reverse approach; calibration in Co-Tuning is necessary; and Co-Tuning fits better for transfer learning than LwF because Co-Tuning explicitly models the category relationship. Adding new layers on top of source logits is a misuse of pre-trained task-specific parameters, and is even inferior to vanilla fine-tuning.

# 6 Why Co-Tuning Works?

We attribute the empirical success of Co-Tuning to the full transfer of pre-trained model and the diversity of categories in pre-trained datasets. Take ImageNet as an example: if the model is pre-trained in ImageNet, the diverse category information in ImageNet makes them capable to compose new classes in a target dataset, enabling Co-Tuning to work even for semantically distinct datasets like Places365 (Zhou et al., 2018) and fine-grained classification datasets like CUB.

For Places365 (a scene recognition dataset), if we sample 100 images per class for training, Fine-tuning achieves Top-1 accuracy 42.49% while Co-Tuning improves it to 43.56%. Although Places365 is distinct to object classification, categories learned in the latter can help scene recognition (e.g. existence of food indicates a "kitchen" scene).

For CUB (a fine-grained classification dataset), target classes are very similar. However, provided that the pre-trained dataset is diverse enough, similar classes can have different source distributions. Take two bird species

| CUB Class | Top 3 Similar ImageNet Class | | |
|---|---|---|---|
| `Crested Auklet` | black swan | oystercatcher | black grouse |
| `Parakeet Auklet` | black grouse | oystercatcher | junco |

`Crested Auklet` and `Parakeet Auklet` as example, top 3 similar ImageNet classes are in the right table to roughly represent their source distributions. Their distributions are similar (both have `black grouse` and `oystercatcher`), but still differ (one has `black swan` while the other has `junco`). In this task, Co-Tuning works for similar classes by supplementing their specific cues.

The assumption of diverse pre-trained datasets also holds in NLP. When we analyze the relationship $p(y_s|y_t)$ in the NER task, where $\mathcal{Y}_t$ is the entity space and $\mathcal{Y}_s$ refers to the vocabulary, we find that words with high probabilities of $p(y_s|y_t = $ "location") are "Germany", "France", "Australia", etc, which corresponds well to the intuition that country names often occur in names of location entities.

Does Co-Tuning help transfer to a dataset with more classes than the pre-trained dataset? We cannot directly answer this question because we find no target datasets with more classes than 1000 classes in the pre-trained ImageNet. However, Co-Tuning works across a wide spectrum of class configurations (COCO-70: 70 classes, Aircraft: 100 classes, Cars: 196 classes, CUB: 200 classes, and Places365: 365 classes). Note that these classes have little overlap with ImageNet classes. Therefore, we hold a positive answer to this question and would like to validate it when a larger dataset occurs.

## 7 Conclusion

In this paper, we propose Co-Tuning, a simple and effective framework for full transfer of pre-trained models. It learns the category relationship and translates target labels into probabilistic source labels. These two labels collaboratively supervise the transfer learning process. Extensive experiments in five tasks validate the efficacy of Co-Tuning across different pre-trained models and dataset sizes.

## Broader Impact

The framework proposed in this paper can be applied to transfer learning scenarios with classification tasks. It can improve the effect of fine-tuning. The broader impact of this paper depends on where fine-tuning is applied. If fine-tuning is used in a suitable way (e.g. train a classifier to automatically recognize if some waste is recyclable or not), then our work can have positive impact in the society. If people use fine-tuning in an evil way (e.g. train a classifier for automatic weapons), then our work can have negative impact in the society. Nevertheless, we hold a positive view of the broader impact of this paper in general.

## Acknowledgments

This work was supported by the National Natural Science Foundation of China (61772299, 71690231), Beijing Nova Program (Z201100006820041), University S&T Innovation Plan by the Ministry of Education of China.We thank anonymous reviewers and meta-reviewers for their constructive reviews.

## Footnotes

[1] A classification dataset constructed from COCO (Lin et al., 2014). See Sec. 5.3 for details.

[2]Notation overload: $f_0$ can refer to a neural network with `softmax` activation (the output is a probability), or without `softmax` activation (the output are logits). The context will make it clear if `softmax` is included.

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
