[Reviews · NeurIPS 2020]

Review 1

Summary and Contributions: In this paper, the authors propose the co-training method for effectively fine-tuning a model pre-trained on a large-scale to a dataset of a specific target application. The hard labels of the target dataset and the probability obtained from the pre-trained model are used for transferring the pre-trained model. The authors have demonstrated that the proposed method is effective when transferring a pre-trained model to several datasets. In particular, the smaller the number of samples of the target data, the more effectively it works.

Strengths: The proposed method is simple and intuitive. The paper is well-written and easy to understand. It seems new that the probability obtained from the pre-trained model is used as a label in fine-tuning. The experimental results are also excellent.

Weaknesses: There should be a more detailed explanation of implementation detail. For example, if the sampling rate is 20%, make sure that the remaining 80% training data is not used at all, or that only the 80% data label is not used. It would be better if the theoretical background is supported. Please provide a more convincing reason for why the proposed method works well. --------------------- After rebuttal ---------------------- The authors gave sufficient answers to my concerns in the rebuttal stage. Thus, I will keep my score.

Correctness: Yes, the author’s claim is correct and easy to understand.

Clarity: Yes, the paper is well written.

Relation to Prior Work: No, it should be discussed. The authors should include more theoretical backgrounds and references to show the difference between the proposed and previous methods.

Reproducibility: Yes

Additional Feedback: There are some minor grammatical errors. Please correct them as below. help of learned category -> help of the learned category pre-trained networks provides -> pre-trained networks provide where () is cross-entropy loss -> where () is a cross-entropy loss output is probability -> output is a probability log likelihood -> log-likelihood with large-scale dataset -> with a large-scale dataset


Review 2

Summary and Contributions: Authors argue that a common practice of fine tuning, which involves dropping the head (output) layer of the source model and replacing it with new head for a target data, is flawed. Heads contain a large number of parameters that can carry valuable knowledge. Authors instead propose to learn the connection between the source classes (thus keeping the source head) and target classes and fine tune using this information.

Strengths: An interesting idea that head source parameters contain valuable information

Weaknesses: It is seems pretty ad hoc. As of now the value of the method is not clear, the method seems overly complicated and is two stage, experiments are not conclusive and not well described, making me question the validity of them Not clear whether target labels used for learning ys->yt connection are excluded from the final evaluation.

Correctness: Not sure that it is correct

Clarity: No, the paper is somewhat hard to follow. No good justification is given to their proposed two stage method

Relation to Prior Work: yes

Reproducibility: No

Additional Feedback: UPDATE: Authors' rebuttal somewhat alleviated my concerns about using the target labels for learning the class connections and about the baselines. I am changing my score up a bit General comments: 1) The process seems like a two step (as opposed learning end to end) - first derive the connection of source and target labels (train a separate network to do this), and then using this connection, train a target model while requiring the output (target labels) to conform to this derived connection. This seems overly complicated. Are both steps happening on the same target dataset?  2) Seems that is only applicable for transferring from large number of source classes to target. Not clear whether it works when number of target classes is larger than number of source classes 3) Authors state that their setting is when source data is not available, but actually their calibration requires the source data. if it is the case, then it would be fair to compare to reweighing techniques that train on both source and target data 4) The first stage (algo 1) itself has hyperparameters already (e.g. g's architecture). 5) Not sure why you have to calibrate the output of the source model - if it was trained with Cross entropy, it should be already calibrated. Alternatively, neural net g should be able to learn the calibration in theory, as long as enough complexity is used . Experiments: 1) A reasonable baseline would just be source model (full) +  one or several new layers for the target. So your target layers are trained on the output of the source model class predictions. Did you compare this with your two stage approach 2) From the algorithm 1 it seems that u are using target labels already, are u using different target labels for the final testing of your results 3) Not clear if the parameters were tuned for other methods, and if yes, on what data. Based on that you said learning rate= 10x original learning rate, I assume you didn't tune the learning rate etc. Also this "hyperparametes .. are selected by the performance on target validation data - are you reporting the final number on the same target validation data? 4) Table 2: is it SD that you report, or it is SE error. If it is SD, then lots of results are not significant Table 3 results are not really conclusive. You should not bolden the results where they are not significant. For example 50% column 83.07 vs 83.43 is not, even assuming you report SE and not SD Minor: please re-read and correct some minor language errors:eg line 121 function retains => is retained, or remains/kept line 203 denotes the rest layers => the rest of the layers line 253 is overwhelming -> ubiquitous or is overwhelmingly used


Review 3

Summary and Contributions: The paper proposes a co-tuning strategy as a replacement for the common fine-tuning technique. The proposed strategy avoid recycles the linear classifier for the source task (often discarded when finetuning) and uses it to provide additional supervision. This is accomplished by 1) learning the relationship between source and task classes as a probabilistic mapping P(Ys|Yt), 2) maintaining two parallel classifiers for source and target classes, and 3) supervising both classifiers using the one-hot label for target classes and the learned probabilistic relationships as a soft label for the source classifier. The proposed method was evaluated on several transfer tasks. For vision tasks, the evaluation was conducted by transferring networks from ImageNet to three medium-sized datasets (Birds, Cars, and Aircraft) and one larger dataset (MS-COCO). For NLP tasks, the method was evaluated by transferring BERT to Named Entity Recognition task. Across all task the proposed approach significantly outperforms the finetuning baseline and three other recent transfer learning algorithms. The gains over prior works are exceptionally large when the target dataset is small, but still provides significant gains in the large data regime.

Strengths: The procedure is simple enough to be used in practice, and provide significant gains over the wide-spread finetuning procedure, as well as, other prior work. The gains are especially large in the small data regime, and still significant in the large data regime. The procedure is also evaluated on a large number of datasets, including both vision and NLP tasks.

Weaknesses: The main weaknesses of this paper are the following (in order of importance). 1. The paper motivation for the proposed transfer procedure is that, in standard finetuning, the classifier for the source classes is discarded (which the paper argues that it's wasteful). However, it is also possible to avoid discarding the source classifier by simply training a classifier for the target task on top of the source classifier. For example, in the case of ImageNet transfer, the target classifier would take the 1000-dimensional logits for ImageNet classes as input, and the whole network would be trained simply to minimize a target classification loss). Although this procedure is not as common as the one described in the paper, it is still used in practice. Thus, it's an important baseline, which is missing in the paper. 2. An obvious limitation of the proposed procedure arises when target classes are semantically distinct from source classes, and thus a reliable model for predicting target classes from source logits cannot be computed. One potential example is when transferring ImageNet pretrained networks to scene recognition (eg Places dataset). However, this limitation has not been addressed. Since this paper relies on source classes for transfer, more attention should have been given to the semantic gap between source and target classes. I believe an experimental study focusing on the transfer performance as a function of this semantic gap is required. 3. The conclusion that the proposed procedure can cooperate with prior work is not supported by Table 4. Although Co-Tuning+X improves upon X in Table 4, the improvements over Co-Tuning alone (i.e. comparing to Table 1) are quite small. 4. L147 says that (2) is a poor estimator of P(Ys|Yt), but no evidence or reference to prior work is given. Would this be true even after the calibration of f0()? It would be great if the authors could add an ablation to Table 5 that uses a calibrated version of (2) as the estimator. 5. While not essential, it would be great if the authors could add a study of the \lambda hyper-parameter (at least in supplementary material). Given the simplicity of the proposed procedure and strong results obtained (especially in the small data regime), the paper has the potential to be accepted at NeurIPS. However, I cannot recommend acceptance due to the above limitations. If the authors adequately address the above concerns, I'm open to reconsidering my position.

Correctness: Method and empirical methodology is sound. See weaknesses for more detail comments on methodology, though.

Clarity: The organization of section 4 can be improved. I found myself having to skip some parts because essential details were only provided later. For example, algorithm 1 mentions a calibration algorithm, but there was no mention of calibration up to that point. Also, I only after reading 4.2, I was able to fully understand section 4.1. This is likely because the procedure is only presented as a algorithm (and thus, motivation for the different components is not provided).

Relation to Prior Work: Relation to prior work is mostly well addressed.

Reproducibility: Yes

Additional Feedback: Post rebuttal comments: Thanks to the authors for taking into account my review. The rebuttal partially addressed my concerns. Given the positive partial results provided, I am optimistic that the authors can add full experiments to the paper. As a result I am increasing my recommendation to a passing score (6). However, I do not fully champion the paper, since I still find my first two weaknesses important, and the results provided in rebuttal are not complete. Detailed post rebuttal comment: Weakness 1) The result provided by the authors partially addresses this concern. I understand that using the classifier logits from the source does not always make sense. In the ImageNet->CUB transfer setting provided by the authors in rebuttal, most of the classes in ImageNet are not related to the birds classes of CUB. So the source logits are not useful, but that is not always the case. In scene recognition, the ImageNet classifier can be used to find objects which can later be used to identify scenes. So, while I understand that reusing the classifier is not always warranted, I disagree that it is a "misuse". I would encourage the authors to add this baseline to their experiments. Weakness 2) The result provided be authors addresses my concern. However, I would like to point out the smaller gains obtained in this setting, compared to the gains obtained in Table 2 for other datasets more similar to ImageNet. I think that adding this to the paper, together with the other baselines, is important. Weaknesses 3 and 4) Fully resolved. No longer a concern of mine. Weakness 5) Not resolved. This was a minor concern, but I still think that a study of any outstanding hyper-parameters is good practice. I'd like to see them in the next iteration of the paper. Clarity) Not resolved. I understand that nothing could be done in rebuttal, but I really hope the authors re-organize and add context to section 4. This was also a minor concern, since I did end up understanding the paper.


Review 4

Summary and Contributions: The paper proposes a new technique for fine-tuning(FT) called co-tuning which also utilizes the task specific layers of the pre-trained model unlike the traditional FT approach which discards the task specific layers(G) and uses a randomly initialized fully connected layer(H) over the bottom layers(F) of the pre-trained model. In co-tuning, in addition to learning F->H, F->G is also utilized under a conditional distribution between the source and target data labels. To learn the category relationships, the authors first calibrate (F->G) on the validation data of the pre-training task(source), then use this to create a dataset and learn a multi-class LR to get the probability of each target class given a source class. They then calibrate this multi-class LR and use Bayes' rule to get the probability of each source class given a target class. Co-tuning trains F->H, using the standard cross-entropy loss, and F->G jointly, the latter being learned using the category probability distribution learnt previously. Only F->H is used during the inference time thereby using co-tuning to improve the learning of the lower layers F. The authors evaluated their approach on 4 computer vision datasets: 3 medium sized and 1 large sized (COCO-70); and 1 Named Entity Recognition NLP dataset by transferring pre-trained models like ResNet-50, DenseNet-121 and BERT. The results show improvements over regularization based FT approaches (whose benefits fade with increasing data size) and can additionally supplement the performance of these methods. The authors also present a small ablation study demonstrating the benefits of using calibration and a multi-class LR for co-tuning.

Strengths: 1) The idea of utilizing the task-specific layers of a pre-trained model to boost fine-tuning is novel and quite interesting. To the best of my knowledge, this has not yet been explored before. 2) The method proposed by the authors is simple, general-enough and easy to understand; and shows consistent performance gains for the computer vision tasks. The baselines used for comparison are adequate to view the advantages of using co-tuning. 3) This work can possibly be beneficial to a large section of the NeurIPS community (both CV and NLP folks) as evident by the wide prevalence of FT pre-trained models.

Weaknesses: [Update after Author Response] Thanks for the author response which has mitigated some minor concerns of mine about specific details of implementation. The only unresolved concern I still have is about the effectivity of co-tuning for simple binary classification tasks in NLP like sentiment analysis, NLI, etc which are widely prevalent in NLP. However, this is a minor concern, and the strong results on the CV tasks and the NER task with a seemingly simple approach is prompting me to increase my score to 7. I would suggest the authors: 1) To dial down the tone of the "uniform applicability of their approach to both CV and NLP tasks" in a future version of their paper as currently, based on the evaluation, I am of the opinion that co-tuning works better in practice for image classification tasks than NLP tasks. 2) To include the references I pointed out in the "Relation to prior work" section for a well rounded related work section. ----------------------------- While the methodology and evaluation conducted shows performance improvements for CV tasks, I have some concerns with regards to the evaluation and applicability to NLP tasks. I am unsure about the authors claim that their technique is applicable to any different domain and hence am assigning this conservative score. I am willing to increase my score if the authors can clarify my concerns: 1) More often than not, since the target task is from a specific domain (say news classification), the probability distribution of the source labels given a target label will be confined to a small overlapping subset of source classes. If multiple target classes have very similar source class distributions, then it is not intuitively clear as to why co-tuning should improve over vanilla-FT. Furthermore, it is not clear if co-tuning is useful when the number of classes for the target task is considerably more than that for the source task? 2) The most common use-case for FT pre-trained transformer models for NLP tasks is for classification tasks like sentiment analysis, NLI, answer sentence selection, etc which are inherently binary classification tasks. Without experiments, a-priori it is unclear if co-tuning might be beneficial for these tasks with so few target classes. 3) The BERT model has been pre-trained with 2 different tasks: masked language modeling (where the target labels are over the vocabulary of size ~30k) and next sentence prediction (where the target label space is discrete: True or False). The authors have only used the MLM task as the source task where the # of classes is close to 30k. If NSP is used as the source task, I am uncertain about any sizable improvements being obtained due to co-tuning. If the authors wanted to consider on a model only pre-trained via the MLM task, they should have used RoBERTa[1] instead of BERT. [1] RoBERTa: A Robustly Optimized BERT Pre-training Approach, Liu et al, 2019

Correctness: For the majority, the claims and methods in the paper are correctly presented and empirically verified. However refer the weaknesses and the questions for more.

Clarity: The paper is well written and easy to understand.

Relation to Prior Work: The paper does a good job of discussing prior work of pre-trained models and fine-tuning practices for computer vision applications. However the paper misses discussing several corresponding ideas in NLP for transfer learning of BERT like models which involve: selective layer freezing, adding a few intermediate parameters, combining embeddings from target specific models, etc. Below are some of the references that the authors might want to include for a more complete discussion of related work in their paper: [2] Parameter-Efficient Transfer Learning for NLP, Houlsby et al, 2019 [3] How to Fine-Tune BERT for Text Classification?, Sun et al, 2019 [4] To Tune or Not To Tune? How About the Best of Both Worlds?, Wang et al, 2019 [5] SimpleTran: Transferring Pre-Trained Sentence Embeddings for Low Resource Text Classification, Garg et al, 2020

Reproducibility: Yes

Additional Feedback: Questions: 1) Algorithm 1 mentions calibrating the pre-trained model on validation data and then using it to learn the category distribution. For the BERT model, what was the development set used by you? 2) [RESOLVED: Authors have indicated that they will include this in the future version] Why have other baseline results not been provided for the NER task in the paper? The paper only mentions the vanilla FT and co-tuning results. 3) [RESOLVED] The standard approach of FT BERT models is using the Adam optimizer with a suitable learning rate. From the paper line #210 it appears that you have used SGD for BERT FT resulting in an evaluation which is not thorough. Can you explain this mismatch? Further can you also explain what you mean by the term "overwhelming" in line #253?

[Author Response · NeurIPS 2020]

# Response to Reviews of "Co-Tuning for Transfer Learning"

We thank all reviewers for their detailed reviews. Although transfer learning has many sub-areas (domain adaptation, fine-tuning, etc), fine-tuning is the most important one that widely affects CV/NLP communities and is a functionality shipped by PyTorch/TensorFlow. However, the major technique is still *feature* fine-tuning (*a.k.a.* **vanilla fine-tuning**), while a simple and more effective alternative is lacking for years. This paper aims to improve over vanilla fine-tuning with *feature and category* **Co-Tuning**, which simply needs *just a few lines of code* (see submitted code). Most reviewers find this paper simple, clearly written with significant results, and appreciate the potential impact on the NeurIPS community. In the following, we respond to common questions first and then to major concerns of each reviewer.

**Common Responses:** **Q1. Details of evaluation and train/val/test split.** Each dataset has a train/test split. For datasets without validation splits, 20% training data are used for validation and the rest 80% training data are used for training. This way, each dataset has a train/**val**/test split. Sampling rate further determines the actually used training data. For example, 30% sampling rate means to use 30% training data for training, leaving the rest 70% training data untouched (both labels and image/text). **Each method has access to the same set of training data.** In Alg. 1, Co-Tuning further splits the sampled training data into secondary train/val (Alg. 1, Line 4) to learn the category relationship. The empirical evaluation is **fair** because **Co-Tuning uses no extra data/labels compared to baselines**.

**Q2. Adding new layers on top of source logits.** Co-Tuning has two heads (Fig. 3): head-1 for target classification (same as vanilla fine-tuning) and head-2 for injecting additional supervision. Adding new layers on top of source logits means to use the output of head-2 for parametric classification, which is actually a misuse. Its accuracy is just **39.26%**, largely inferior to Co-Tuning (**52.58%**) and even inferior to vanilla fine-tuning (**45.25%**) in the setup of Table 5.

**Q3. Does Co-Tuning help transfer to a dataset with more classes than the pre-trained dataset?** We cannot directly answer this question because we find no target datasets with more classes than 1000 classes in the pre-trained ImageNet. However, co-tuning works across a wide spectrum of class configurations (COCO-70: 70 classes, Aircraft: 100 classes, Cars: 196 classes, CUB: 200 classes, and Places365 in R3/Q1: 365 classes). Note that these classes have **little overlap** with ImageNet classes. Therefore, we hold a positive answer to this question.

**Response to R1:** In fact, the vanilla fine-tuning still lacks a complete theory, not to mention Co-Tuning. Research of deep learning algorithms usually precedes the theories. We are actively seeking theoretical support for Co-Tuning.

**Response to R2:** Pre-train/fine-tune itself is two-stage. Co-Tuning is acceptable in this regard because it works.

**Q1. Why is Co-Tuning simple?** The core of Co-Tuning is to add supervision by the pre-trained task-specific classifier (**a few lines of code**). $g$'s architecture can be fixed as one layer (*i.e.* Softmax regression), which is simple but works well for all datasets. While calibration is important as stated in [On Calibration of Modern Neural Networks, ICML 2017] (Citations 750), Co-Tuning without calibration (thereby without source data) already works well (Table 5). Even compared with re-weighting method [Learning to Transfer Learn: Reinforcement Learning-Based Selection for Adaptive Transfer Learning, ECCV 2020] which achieves 49.62% in Table 5 setup, Co-Tuning (52.58%) is better because it focuses on target data while the former may be biased by source data. The simplicity is agreed by the R1/3/4.

**Q2. Hyper-parameter tuning details.** All methods (both baselines and Co-Tuning) follow the same hyper-parameter tuning procedure. Each dataset has a validation split (common responses Q1), on which hyper-parameters are tuned. The learning rate ratio of new layers to pre-trained layers is fixed to 10 (a standard choice in fine-tuning), but the learning rate of pre-trained layers is tuned on validation data. We treat all methods the same and present a **fair comparison.**

**Q3. Significance of results.** We report standard deviations. Many gains exceed three times of standard deviation, and the rest reviewers all agree the results are significant. We will un-bold several insignificant results.

**Response to R3:** **Q1. Results on Places365 (semantically distinct dataset).** We sample 100 images per class for training. Fine-tuning achieves Top-1 accuracy **42.49%** while Co-Tuning improves it to **43.56%**. Although Places365 is distinct to object classification, categories learned in the latter can help scene recognition (*e.g.* existence of food indicates a "kitchen" scene). Co-Tuning may work in a wide spectrum of semantic gap between datasets.

**Q2. Improvement of Co-Tuning + baseline over Co-Tuning is small.** We will remove Table 4 and this argument. We will instead focus on improving Co-Tuning itself, bringing the community a strong alternative to the vanilla fine-tuning.

**Q3. Use a calibrated version of Eq. (2) as the estimator.** Even after calibrating the source logits $f0()$, its accuracy (**47.52%**) is inferior to Co-Tuning (**52.58%**) in the setup of Table 5. We will add these results in the paper.

**Response to R4:** Many thanks for your appreciation. Due to space limits, minor concerns will be addressed in a revision (more baselines, more tasks like NLI, etc). Implementation will be crystal clear since we will open-source.

**Q1. Why Co-Tuning works when target classes are similar.** *Even if target classes are similar, they have different source distributions provided that the pre-trained dataset is diverse enough.* Take two bird species "Crested Auklet" and "Parakeet

| CUB Class | Top 3 Similar ImageNet Class | | |
|---|---|---|---|
| Crested Auklet | black swan | oystercatcher | black grouse |
| Parakeet Auklet | black grouse | oystercatcher | junco |

Auklet" as example, top 3 similar ImageNet classes are in the right table to roughly represent their source distributions. Their distributions are similar (both have "black grouse" and "oystercatcher"), but still differ (one has "black swan" while the other has "junco"). In summary, Co-Tuning works for similar classes by supplementing their specific cues.

**Q2. NLP experiments.** Because NSP has binary outputs, transferring its categories makes less sense. The optimizer is Adam following the example code in Transformers library provided by Hugging Face. We will update Line 210.

[Meta-Review · NeurIPS 2020]

This paper presents a simple method which seems to work well in practice. Some reviewers would have preferred to see more discussion on limitations of the method. However, the contribution was deemed clear enough without this discussion, because of the intuitive, novel take on the popular fine-tuning task, as well as the strong performance demonstrated on popular vision tasks. Overall, the paper is expected to be of interest to the community.